# Antinociceptive effect of *Equisetum arvense* extract on the stomatitis hamster model

**Fumie Shiba**[ID][1]*, **Shiiko Maekawara**[2], **Atsuko Inoue**[3,4], **Koji Ohta**[5], **Mutsumi Miyauchi**[ID][1,6]*

**1** Collaborative Research Laboratory of Oral Inflammation Regulation, Graduate School of Biomedical and Health Sciences, Hiroshima University, Hiroshima, Japan, **2** Faculty of Dentistry, Hiroshima University, Hiroshima, Japan, **3** Department of Pharmacotherapeutics, Graduate School of Pharmacy and Pharmaceutical Sciences, Fukuyama University, Fukuyama, Hiroshima, Japan, **4** Department of Pharmacotherapeutics, Fukuyama University, Fukuyama, Hiroshima, Japan, **5** Department of Public Oral Health, Program of Oral Health Sciences, Graduate School of Biomedical and Health Sciences, Hiroshima University, Hiroshima, Japan, **6** Department of Oral and Maxillofacial Pathobiology, Graduate School of Biomedical and Health Sciences, Hiroshima University, Hiroshima, Japan

* mmiya@hiroshima-u.ac.jp (MM); fshiba@hiroshima-u.ac.jp (FS)

## Abstract

Recurrent aphthous stomatitis leads to ulcers that cause severe pain, which is a substantial burden on patients. *Equisetum arvense* extract (EA) is a crude drug that promotes wound healing of mucous membranes caused by perineal incision during childbirth and alleviates pain. Here, we elucidated the effects of EA on wound healing and pain in a stomatitis hamster model. After stomatitis induction, two different EA doses were continuously applied to the wound area through the intramucosal injection of acetic acid into the cheek pouch (stomatitis/100*EA group and stomatitis/EA group). The body weight and wound area were measured over time, and histological evaluation was performed after stomatitis induction. The wound area was harvested 10 h after stomatitis induction, and gene expression associated with pain and inflammation was analyzed using qPCR. The dorsal root ganglia of the rat spinal cord were isolated, dispersed, and cultured to examine the inhibitory effect of EA on the $K^+$-evoked release of neurotransmitter substance P. In the stomatitis/100*EA group, a significant reduction in wound size was observed compared with the stomatitis/physiological saline (PS) group, and the weight gain rate was considerably higher. The stomatitis/EA group revealed similar histological changes in the wound and wound size as the stomatitis/PS group; however, the weight gain rate was considerably higher on day 7. The stomatitis/EA group suppressed the expression of inflammatory cytokine mRNA, such as *Tnf-α* and *Il-6*, and *Cox-2* mRNA in the wound area compared with the stomatitis/PS group. EA treatment reduced the upregulated substance P release from the dorsal root ganglia following high-concentration $K^+$ stimulation. EA alleviates pain in a stomatitis model by suppressing inflammatory cytokine expression in the wound area and substance P release from primary sensory neurons. Therefore, using oral care products containing EA is expected to suppress stomatitis pain.

**Data Availability Statement:** All relevant data are within the manuscript and its Supporting Information files.

**Funding:** F.S. and M.M. belonged to a collaborative research laboratory of Earth Corporation (https://

corp.earth.jp/jp/index.html) and Hiroshima University, which used joint research funds to conduct this study. The funders had no role in study design, data collection and analysis, decision to publish, or preparation of the manuscript.

**Competing interests:** F.S. and M.M. belonged to a collaborative research laboratory of Earth Corporation (https://corp.earth.jp/en/index.html) and Hiroshima University, which used joint research funds to conduct this study. This does not alter our adherence to PLOS ONE policies on sharing data and materials.

## Introduction

Stomatitis is a prevalent oral disease that manifests in various forms, such as aphthous, catarrhal, and recurrent aphthous stomatitis (RAS) and stomatitis due to chemotherapy side effects and candida infection [1–5]. The different types of stomatitis have different pathogeneses; however, they all impact patients' quality of life and interfere with activities of daily living since they all cause pain due to erosion and/or ulcer formation on the mucosal epithelium.

RAS is the most common stomatitis type, and its prevalence in the general population is reportedly 20% [6]. The etiology of RAS lesions remains unclear; however, several local, systemic, immunological, genetic, allergic, nutritional, and microbial factors have been proposed as causes. Additionally, a few immunosuppressive drugs are associated with severe aphthous-like stomatitis [7,8]. Moreover, mucosal injury [9] initiates RAS and can develop iatrogenically because of orthodontic appliances and unsuitable dentures [10–12].

Therefore, managing RAS is challenging; currently, no definitive treatment exists to control the symptoms. Severe pain caused by ulcers is a substantial burden on patients with RAS. Corticosteroids are often used as a symptomatic treatment for RAS. However, alternative substances that improve stomatitis symptoms and promote healing are required because steroids can lead to infection and immune depression. Thus, natural anti-inflammatory substances from medicinal herbs are promising agents for RAS management.

The genus *Equisetum* is widely distributed worldwide and is considered one of the oldest extant genera on Earth [13–16]. *Equisetum arvense* is the most popular species for medicinal uses. *E. arvense* extract (EA) has various potential pharmacological properties, including anti-inflammatory, anti-osteoclastogenesis, and antioxidant effects [17–22]. Additionally, EA promotes wound healing of the mucous membranes caused by perineal incision during childbirth and alleviates pain [21]. Moreover, EA considerably promotes skin wound healing in rats [23]. EA inhibits nociceptive pain in nociception models through mechanisms that differ from those in the opioid system [20]. However, its effects on stomatitis-related pain have not been elucidated.

Therefore, in this study, we investigated the effect of EA on stomatitis healing and pain by evaluating its healing-promoting effects on stomatitis based on cure rate, histology, and gene expression of pain-related factors at the stomatitis site using a stomatitis hamster model. Furthermore, we examined the inhibitory effects of EA on substance P (SP) release from the cultured dorsal root ganglia (DRG) cells.

## Materials and methods

### Reagents

EA (extracted from whole *E. arvense* using a 50% 1,3-butylene glycol solution) was purchased from Maruzen Pharmaceuticals Co., Ltd. (Hiroshima, Japan), and acetic acid was purchased from Sigma-Aldrich (St. Louis, MO, USA). The acetic acid solution used in the experiments was prepared using saline (Otsuka Normal Saline, Otsuka Pharmaceutical Factory, Inc., Tokyo, Japan).

### Materials

Dulbecco's modified Eagle's medium was purchased from Nissui Pharmaceutical Co. (Tokyo, Japan). Horse serum and penicillin/streptomycin were purchased from Gibco BRL (Gaithersburg, MD, USA). Trypsin 2.5% was obtained from Invitrogen (Burlington, Ontario, Canada), and the mouse laminin was purchased from Upstate Biotechnology (Lake Placid, NY, USA). Collagenase and polyethylenemine were purchased from Sigma

Chemical Co. (St Louis, MO, USA). Mouse nerve growth factor (NGF) was purchased from Promega Co. (Madison, WI, USA).

## Animals

All experiments were approved by the Ethics Committee for Animal Experimentation of Hiroshima University (Permit Number: A20-45) and were performed according to the "Guidelines for the Care and Use of Laboratory Animals" established by Hiroshima University, following the ARRIVE [24] and AVMA guidelines. Overall, 52 5-week-old male golden Syrian hamsters (*Mesocricetus auratus*) weighing 95.6 ± 8.4 g (81.6–115.9 g) (S1 Table) (Charles River Japan, Inc., Yokohama, Japan) were housed in a specific pathogen-free facility in 12-h light-dark cycles with *ad libitum* access to water and food and kept at a constant ambient temperature and humidity (22˚C and 50 ± 5% relative humidity). The maximum number of animals per cage was three. Therefore, each group was kept in its cage to prevent grooming contact with other groups.

The hamsters were anesthetized during the experiment through a peritoneal injection of a mixture of three anesthetics (3 mL/kg/animal): medetomidine hydrochloride (0.03 mg/mL; Dorbene® vet, Kyoritsuseiyaku Co., Tokyo, Japan), midazolam (0.4 mg/mL; Midazolam Injection 10 mg, Sandoz K.K., Tokyo, Japan), and butorphanol tartrate (0.5 mg/mL; Vetorphale®, Meiji Seika Pharma Co., Ltd., Tokyo, Japan), which were adjusted with saline (Otsuka Normal Saline). This anesthetic had been used in our previous animal study [18]. All efforts were made to minimize animal suffering. The hamsters were fixed to their backs on an experimental stand. Furthermore, all animals were sacrificed after the experiment using $CO_2$ gas, and all data were analyzed with each animal represented as n=1.

## Stomatitis hamster model induction through acetic acid injection

The stomatitis hamster model was established based on previous studies [25,26]. A total of thirty-seven hamsters were divided into three groups of ten (stomatitis/PS, stomatitis/EA and stomatitis/100*EA groups) and one group of seven (healthy group). Under anesthesia, the right cheek pouches of the hamsters were extended outside the oral cavity. Acetic acid solution (30 µL; 10%) was injected into the submucosal connective tissue of the cheek pouch using a microsyringe (BD Low-Dose TM U-100 Insulin Syringe, 0.5 mL, 30 G, 8 mm, Nippon Becton Dickinson Co., Ltd. Tokyo, Japan), excluding the healthy group. Subsequently, the cheek pouch was placed back into the oral cavity, and the animals were returned to the cage. Starting from the day after stomatitis induction, 50 µL of EA (15 µg/mL or 1500 µg/mL) was applied once daily for seven consecutive days (stomatitis/EA or stomatitis/100*EA group), and the wound area (boundary-relatively clear round white elevated lesion including ulcer) was measured. Physiological saline (PS) was administered to the lesions in the control group (stomatitis/PS group). Furthermore, the long and short diameters of the wounds were measured using a microcaliper, and the wound size was determined by multiplying the two values. The healing efficacy of each specimen was evaluated by comparing cure rates over time, which was calculated using the following formula:

$$\text{Cure rate (\%)} = \frac{\text{wound size of d1 mesaurement} - \text{wound size of dn measurement}}{\text{wound size of d1 measurement}} \times 100$$

Three hamsters from groups stomatitis/PS, stomatitis/EA, and stomatitis/100*EA were sacrificed on the fourth day after the intramucosal injection of acetic acid solution into the cheek pouch, while the remaining hamsters from groups healthy, stomatitis/PS, stomatitis/EA, and

stomatitis/100*EA were sacrificed on the seventh day after the same injection; the cheek pouches were collected, and the tissue sections were prepared for histological evaluation.

The following formula was used to determine the weight gain from the day after acetic acid injection:

$$\text{Percent weight gain (\%)} = [(\text{dn}-\text{d1})/\text{d1}] \times 100$$

where d1 is the hamster's weight on the day after acetic acid injection, and dn is the weight recorded n days after acetic acid injection.

## Histological analysis

Tissue samples were collected 4 and 7 days after stomatitis induction and fixed with 4% para-formaldehyde. Furthermore, paraffin tissue blocks were prepared, and 4.5 μm thick sections were cut using a microtome. The sections were stained with hematoxylin and eosin for routine histological evaluation, or with Masson's trichrome to observe collagen fibers under a microscope (S1 Fig).

## Genetic analysis through quantitative polymerase chain reaction

Fifteen hamsters were divided into three groups (healthy, stomatitis/PS, and stomatitis/EA) of five hamsters each. Acetic acid solution (30 μL; 10%) was intramucosally injected into the cheek pouches of 10 hamsters, excluding the healthy group. Furthermore, 50 μL of EA (15 μL/mL) or PS (control) was immediately administered before and after the intradermal injection. After 10 h, the tissues around the injection area were collected from all hamsters, and the total RNA was extracted and purified from the hamster tissues using TRIzol reagent (Invitrogen, Tokyo, Japan) after manually grinding the tissue using a sterile grinding stick. After complementary DNA synthesis using the ReverTra Ace® (Toyobo Co., Ltd., Osaka, Japan) following the manufacturer's protocol, the Applied Biosystems® StepOne™ Real-time PCR System (Thermo Fisher Scientific, MA, USA) was used with FastStart Essential DNA Green Master (Roche, Basel, Switzerland) according to the manufacturer's instructions. Reactions were conducted for up to 45 cycles with denaturing at 95°C, annealing at 55°C, and extension at 72°C.

The primers used for this system were as follows: hamster tumor necrosis factor (*Tnf*)-α, 5′-CTCCTTCCTGCTTGTGGGAG-3′ (sense) and 5′-GAGCCGATGATAGGGTTGGG-3′ (antisense); hamster cyclooxygenase-2 (*Cox-2*), 5′- ATGACTGCCCAACTCCCTTG-3′ (sense) and 5′- ACACCTCTCCACCAATGACC-3′ (antisense); hamster interleukin-6 (*Il-6*), 5′-TCTTCTTGGGACTGCTGCTG-3′ (sense) and 5′-TGTTCGTCACAAACTCCAGGT-3′ (antisense); hamster bradykinin receptor B1 (*Bdkrb1*), 5′-CCCTCTAACCAAAGCCAGCA-3′ (sense) and 5′-GCAGGCAGAGATGTTCAGGT-3′ (antisense); hamster glyceraldehyde-3-phosphate dehydrogenase (*Gapdh*) as an internal standard, 5′-ACAGTCAAGGCTGAGAACGG-3′ (sense) and 5′-CAGGCGACATGTGAGATCCA-3′ (antisense). The quantitation cycle (Cq) method was used, and the relative transcript number of the target gene was normalized to that of *Gapdh* using the $2^{-\Delta\Delta Cq}$ method [27].

## Isolating and culturing the DRG neurons

The Animal Care Committee of Hiroshima University approved the experimental protocol (Permit Number: A23-49). The protocol of Inoue et al. and Tang et al. was used as a reference to conduct the experiment [28,29]. Rat DRG were removed from adult Wistar rats (6–9 weeks) and dissociated into single isolated cells through an enzyme treatment of 0.125% collagenase for 90 min (twice), followed by 0.25% trypsin for 30 min at 37°C. The cells (10 DRG/φ35 mm dish) were plated on polyethyleneimine and laminin-coated dishes and incubated in

Dulbecco's modified Eagle's medium containing 10% heat-inactivated horse serum (Gibco™ Horse Serum, heat-inactivated, New Zealand origin, Thermo Fisher Scientific Inc.), 1% penicillin/streptomycin, and 30 ng/mL nerve growth factor (*NGF*, 2.5S, Murine, *#G514A*, Promega). The cultures were maintained at 37°C in a water-saturated atmosphere with 5% $CO_2$ for 5 days before the experiment. The neurons demonstrated globular cell bodies with extended axonal processes on day 5 of culture.

In the SP-release experiments, the cultured cells were moved into Krebs-HEPES buffer (solution of 100 mM of sodium chloride [NaCl], 4.5 mM of potassium chloride [KCl], 2 mM of calcium chloride [$CaCl_2$], 1.2 mM of magnesium sulfate [$MgSO_4$], 1.2 mM of monopotassium phosphate [$KH_2PO_4$], 25 mM of sodium bicarbonate [$NaHCO_3$], 11.7 mM of glucose, and 10 mM of 4- (2-hydroxyethyl)-1-piperazineethanesulfonic acid [HEPES], at pH 7.4) and treated at 37°C for 10 min, and subsequently replaced with a high-potassium (100 mM of $K^+$) Krebs-HEPES buffer (solution of 15.7 mM of NaCl, 98.8 mM of KCl, 2 mM of $CaCl_2$, 1.2 mM of $MgSO_4$, 1.2 mM of $KH_2PO_4$, 25 mM of $NaHCO_3$, 11.7 mM of glucose, and 10 mM of HEPES at pH 7.4) at 37°C for 10 min.

## Measuring the SP content

The Krebs-HEPES and high-potassium Krebs-HEPES buffers were collected after each incubation and used to measure the SP content. The enzyme-linked immunosorbent assay for SP was performed according to the protocol of the Parameter™ Substance P (R&D Systems, Inc., Minneapolis, MN, USA) to measure the SP content in the Krebs-HEPES buffer. The SP release ratio, which represents the ratio of evoked SP release to spontaneous SP release, was calculated using the following formula.

$$\text{SP release ratio} = \frac{\text{SP content in high–potassium Krebs–HEPES buffers (pg/mL)}}{\text{SP content in Krebs–HEPES buffers (pg/mL)}}$$

## Statistical analysis

Results are reported as mean ± standard deviation. Intergroup differences were compared using the Tukey–Kramer multiple comparison test, which was conducted using the multcomp package in R [30]. Statistical significance was set at $p < 0.05$, $p < 0.01$, and $p < 0.001$.

## Results

### Macroscopic findings

We replicated human-like stomatitis, characterized by notable inflammatory cell infiltration at the ulcer and its base, to evaluate the effects of EA on stomatitis following the protocol outlined in a previous study by Katayama et al. [26]. We induced a demarcated circular white lesion with ulcer formation (termed "wound") through an intramucosal acetic acid injection into the cheek pouch.

Using this stomatitis hamster model, the healing-promoting effect of EA on stomatitis was evaluated based on the changes in wound size over time. One day after stomatitis induction, a circular, white, elevated lesion (wound) was observed in the cheek pouch, and the wound size gradually decreased over time. Macroscopically, the difference in the wound size between the stomatitis/EA and stomatitis/PS groups was insignificant. However, the wound size in the stomatitis/100*EA group significantly decreased from 3 days after the start of the experiment

compared with that in the stomatitis/PS group. The results indicate that 15 μg/mL of EA did not promote healing; however, 1500 μg/mL of EA had promoting effects (Fig 1A and 1B).

## Histology of the stomatitis hamster model

Subsequently, we evaluated the healing-promoting effect of EA on the wounds by comparing the histological findings of the wounds on days 4 and 7 after PS or EA was applied daily to the wound. The healthy cheek pouch comprises ① flat and thin, regular keratinized stratified squamous epithelium, ② dense connective tissue just below the epithelium, ③ muscle fibers, and ④ loose connective tissue. However, no inflammatory cell infiltration was observed in any of the layers (Fig 2A).

Furthermore, we confirmed the pathological findings of the wounds in the stomatitis/PS group on day 4. The covering epithelium was necrotic and desquamated from the underlining connective tissue to form an ulcer (▼). Severe neutrophil infiltration was observed in the subepithelial connective, muscular tissues, and the upper part of loose connective tissue (fourth layer), indicating acute-phase inflammation. Additionally, the surfaces of the ulcer and necrotic epithelium were covered with purulent exudate in large amounts (Fig 2B and 2B'). These pathological findings were observed in the stomatitis/EA group (Fig 2C and 2C'). In the epithelial covering of the stomatitis/100*EA group, the covering epithelium was necrotic, and it had peeled off from the underlying connective tissue to form an ulcer (▼). Strong neutrophil

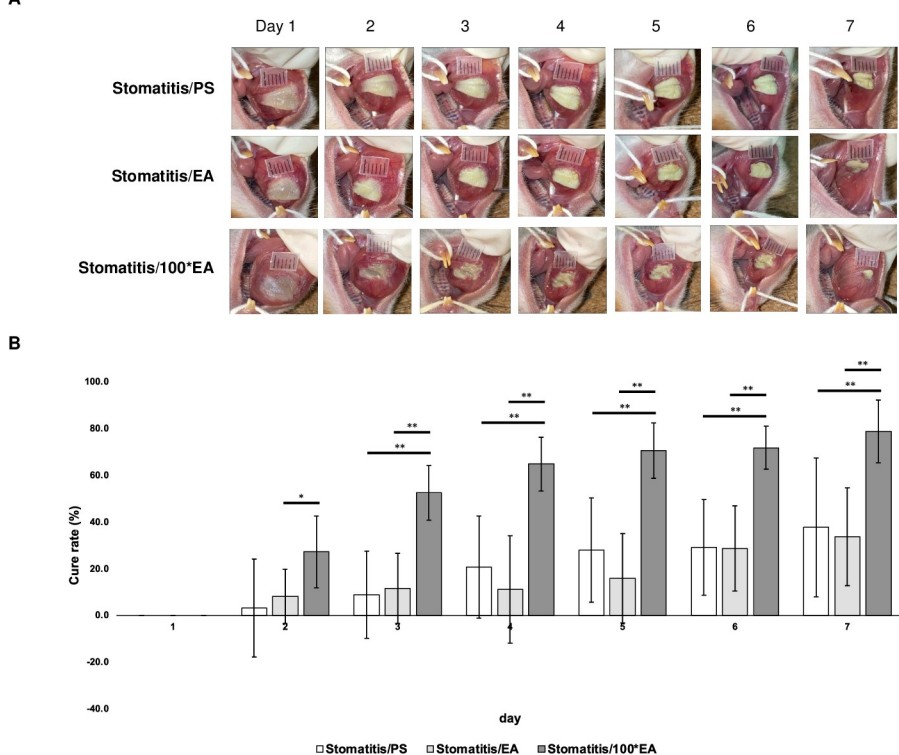

**Fig 1. Cure rate change over time.** Comparison of the cure rates between groups treated with physiological saline (PS) or *Equisetum arvense* extract (EA) daily after inducing stomatitis. (A) Macroscopic changes in the wounds over time for each group. Images were taken with iPhone SE (Apple Inc., CA, USA). (B) The healing efficacy of each animal was evaluated by comparing cure rates over time. Each point represents the mean for seven animals ± standard deviation, and the differences between the three groups were compared using the Tukey–Kramer multiple comparison test, * $p < 0.05$ and ** $p < 0.005$.

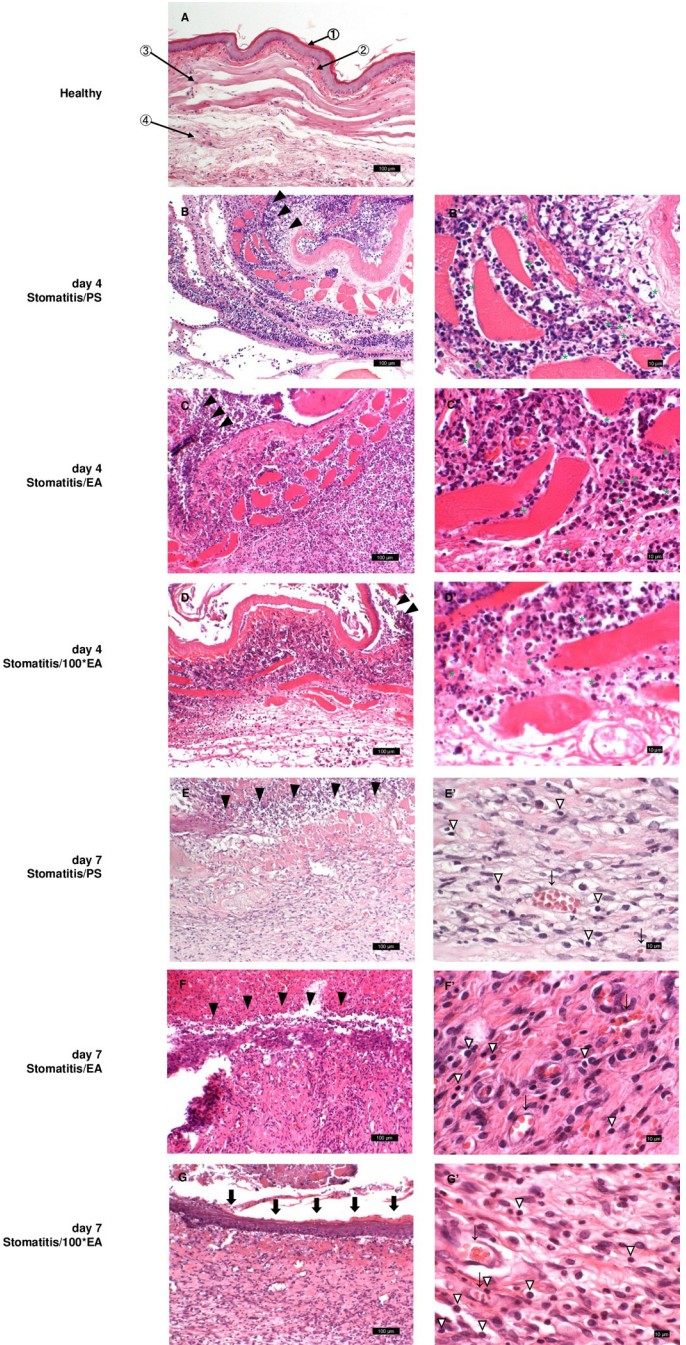

**Fig 2. Histology of the stomatitis hamster model (hematoxylin and eosin staining).** Histological findings on days 4 (B, C, and D) and 7 (E, F, and G) after stomatitis induction. A illustrates a healthy negative control group. B and E represent the stomatitis/PS-positive control group, C and F represent the stomatitis/EA group, and D and G represent the stomatitis/100*EA group. A–G are under low magnification (scale bars: 100 μm); B'–G' are under high magnification (scale bars: 10 μm). (A) Healthy: The cheek pouch comprises ① flat and thin, regular keratinized stratified squamous epithelium, ② dense submucosal connective tissue, ③ a muscular layer, and ④ loose connective tissue. No inflammatory cell infiltration was observed in any of the areas. (B, C, D) Four days after acetic acid-induced stomatitis. Ulcer (▼) is formed at the injection area of the pouch mucosa, and severe neutrophil infiltration (*) was observed in the edematous underlying muscle and connective tissues. (E, F, G) Seven days after acetic acid-induced stomatitis. The epithelial mucosal surface remained ulcerated; however, neutrophil infiltration was restricted to the ulcer floor. Granulation tissue with blood vessels (↓) and chronic inflammatory cell infiltration (▽) proliferate in the deeper portion of the ulcer base, indicating a transition to the healing phase (E, F). Adjacent to the newly formed mucosal epithelium (⬇), granulation tissue was seen (G).

infiltration in the subepithelial connective tissue was still observed; however, it was limited to the muscular layer (Fig 2D and 2D').

Furthermore, the pathological findings of the wound in the stomatitis/PS group on day 7 demonstrated that the mucosal surface was still ulcerated (▼) and covered by the exudate. However, several lymphocytes and plasma cells were observed in the underlying muscular tissue. Additionally, granulation tissue with blood vessels and chronic inflammatory cell infiltration proliferated in the deeper areas of the ulcer base, indicating a transition to the healing phase (Fig 2E and 2E'). The pathological findings in the stomatitis/EA group (Fig 2F and 2F') were similar to those in the stomatitis/PS group, indicating no healing-promoting effect of 15 μg/mL of EA on stomatitis. Conversely, in the stomatitis/100*EA group on day 7, new mucosal epithelium (⬇) was formed under the purulent exudate, covering the wound surface and granulation tissue with chronic inflammatory cell infiltration observed just beneath the area of newly formed covering epithelium (Fig 2G and 2G'), indicating that 1500 μg/mL EA promotes the healing of stomatitis.

Fig 3 reveals the connective tissue condition of the superficial area of the wound in each group on day 7 using Masson's trichrome staining. In the healthy cheek pouch, regularly arranged dense collagen fibers and many fibroblasts (▽) were observed in the submucosal connective tissue (Fig 3A and 3A'). We confirmed that collagen fibers in the wound surface area of the Stomatitis/PS and Stomatitis/EA groups were destroyed by edema and inflammatory cell infiltration, and regular arrangement of spindle fibroblasts was not observed (Fig 3B, 3B', 3C and 3C'). Conversely, in the Stomatitis/100*EA group, the granulation tissue formed below the regenerated mucosal epithelium showed dense collagen fibers with many spindle-shaped fibroblasts (▽) (Fig 3D and 3D'). These histopathological findings indicate that 1500 μg/mL EA promotes the healing of stomatitis.

**Recovery of weight gain with EA.** We noticed that the healthy group, the stomatitis/EA group, and the stomatitis/100*EA group had more food left in their cheek pouches than the stomatitis/PS group. This indicates that the stomatitis/EA group can intake more food despite having the same level of damage as the stomatitis/PS group, suggesting that a low concentration of EA without wound healing effect has an analgesic effect on stomatitis wounds with ulcers. Therefore, we decided to evaluate the rate of weight gain in each group as an indicator of pain.

In the stomatitis/PS group, a significant decrease in weight gain was observed from the fourth day onwards compared to the healthy group (Fig 4). This suggests that the significant decrease in the rate of weight gain in the stomatitis/PS group is possibly due to the severe pain caused by ulcer formation at the site of the stomatitis.

However, in the group that applied different doses of EA to the stomatitis (Stomatitis/EA, Stomatitis/100*EA), there was no significant difference compared with the healthy group at any time point, indicating that EA-treated animals could take enough food despite having an ulcer. In addition, compared with the stomatitis/PS group, the significantly higher weight gain in the stomatitis/100*EA and stomatitis/EA groups was observed from days 3 and 7, respectively. A low dose of EA (stomatitis/EA group) without wound healing effect macroscopically and microscopically showed a similar level of weight gain in healthy controls, suggesting that EA may have an analgesic effect. These results indicate that continuously applying EA can control the pain caused by stomatitis ulcers.

## Analgesic effects of EA on stomatitis

We performed a quantitative polymerase chain reaction (qPCR) to identify the changes in the expression of genes involved in inflammation and pain in the wound area. The inflammation-

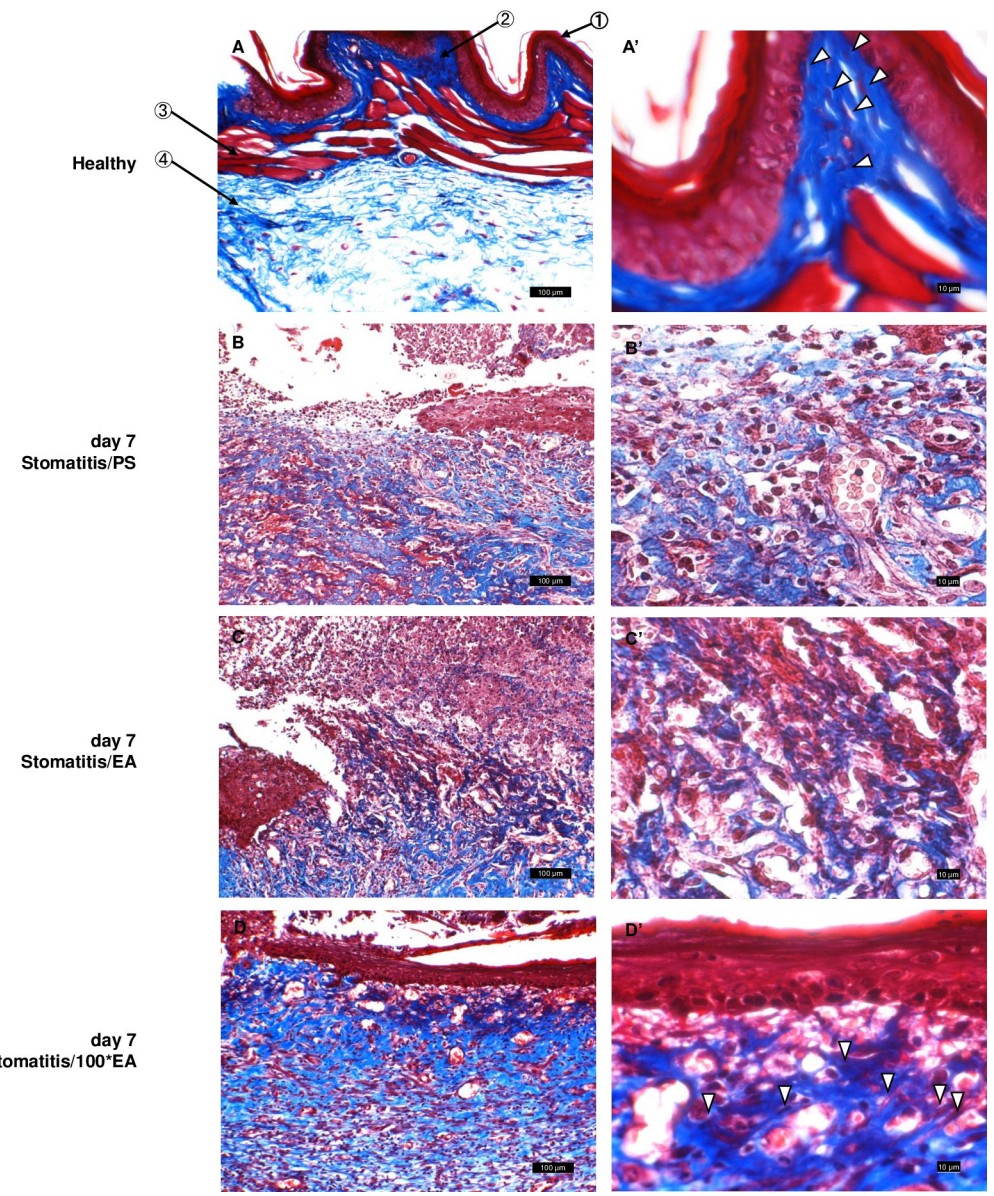

**Fig 3. Histology of the stomatitis hamster model (Masson's trichrome staining).** Histological findings on day 7 after stomatitis induction. A illustrates a healthy negative control group. B represents the stomatitis/PS-positive control group, C represents the stomatitis/EA group, and D represents the stomatitis/100*EA group. A–D are under low magnification (scale bars: 100 µm); B'–D' are under high magnification (scale bars: 10 µm). (A) Healthy: Dens collagen fibers and many fibroblasts (▽) were observed in the subepithelial connective tissues. (B, C) Stomatitis/PS and Stomatitis/EA groups: Destruction of collagen fibers was evident. (D) stomatitis/100*EA group: Abundant collagen fibers were confirmed in the granulation tissue formed directly below the newly formed mucosal epithelium. Moreover, many spindle fibroblasts (▽) were observed.

related genes included *Tnf-α* and *Il-6* mRNA and pain-related genes included *Cox-2* and *Bdkrb1* mRNA. At 10 h after stomatitis induction, *Cox-2* and *Tnf-α* mRNA upregulation observed in the stomatitis/PS group was considerably suppressed in the stomatitis/EA group (Fig 5). *Il-6* mRNA was significantly upregulated in the stomatitis/PS group; however, it decreased in the stomatitis/EA group ($p = 0.0596$). Conversely, *Bdkrb1* mRNA expression did not differ between the stomatitis/EA and stomatitis/PS groups.

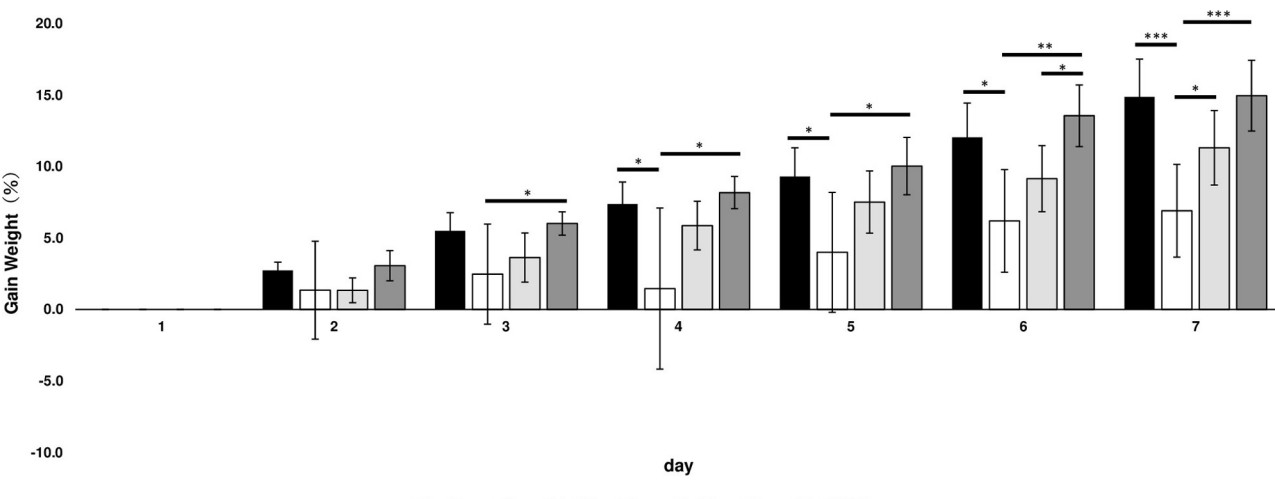

**Fig 4. Comparing the groups regarding animals that gained weight (%).** Comparison of body weight between groups treated with PS (stomatitis/PS) or EA (stomatitis/EA) or 100*EA (stomatitis/100*EA) daily after stomatitis induction. The percentage weight gain for each group was calculated based on the body weight 1 day after stomatitis induction. Each point represents the mean for seven animals ± standard deviation. Tukey–Kramer multiple comparison test, * $p < 0.05$, ** $p < 0.001$ and *** $p < 0.0001$. Healthy: non-stomatitis with PS application, Stomatitis/PS: acetic acid-induced stomatitis with PS application, Stomatitis/EA: acetic acid-induced stomatitis with 15 µg/mL of EA solution application, Stomatitis/100*EA: acetic acid-induced stomatitis with 1500 µg/mL of EA solution application.

## EA suppresses the release of SP from the primary sensory neurons

Cultured DRG cells were used to examine the inhibitory effects of EA on SP release. Cultured DRG cells contain primary sensory neurons and non-neuronal cells around the neurons, where SP is biosynthesized and released in response to noxious stimuli. The results revealed that EA markedly inhibited the $K^+$-evoked release of SP in cultured DRG cells compared with the control group (Fig 6).

## Discussion

We demonstrated that EA promotes the healing of acetic acid-induced ulcers and considerably suppresses SP release and *Cox-2* mRNA expression in an acetic acid-induced wound area, thereby alleviating pain in a stomatitis model. EA exhibits analgesic effects differently from opioids in a chemical pain model in rats using behavioral evaluation; however, the mechanisms of these effects are unclear [20]. To our knowledge, the current study is the first to demonstrate that EA might uniquely reduce stomatitis pain.

In the current study, hamsters were employed as an animal model. Their immune responses are widely recognized as comparable to those of humans, making them suitable for various disease models [31–39]. Specifically, Syrian hamsters are frequently utilized in research on stomatitis [26,40]. Thus, this evidence supports the use of hamsters as an excellent animal model for studying stomatitis.

We used the acetic acid-induced stomatitis hamster model established by Katayama et al. [26] to confirm the analgesic effect of EA on stomatitis. We confirmed histologically that acute stomatitis with ulcer formation could be induced at the acetic acid injection site, and *Cox-2*, *Il-6*, and *Tnf-α* mRNA expression was markedly increased 10 h after the injection. Furthermore, we observed a substantial decrease in the rate of body weight gain, which may be associated with the decreased food intake in the stomatitis model. Based on these findings, we successfully established a stomatitis model for inflammatory pain.

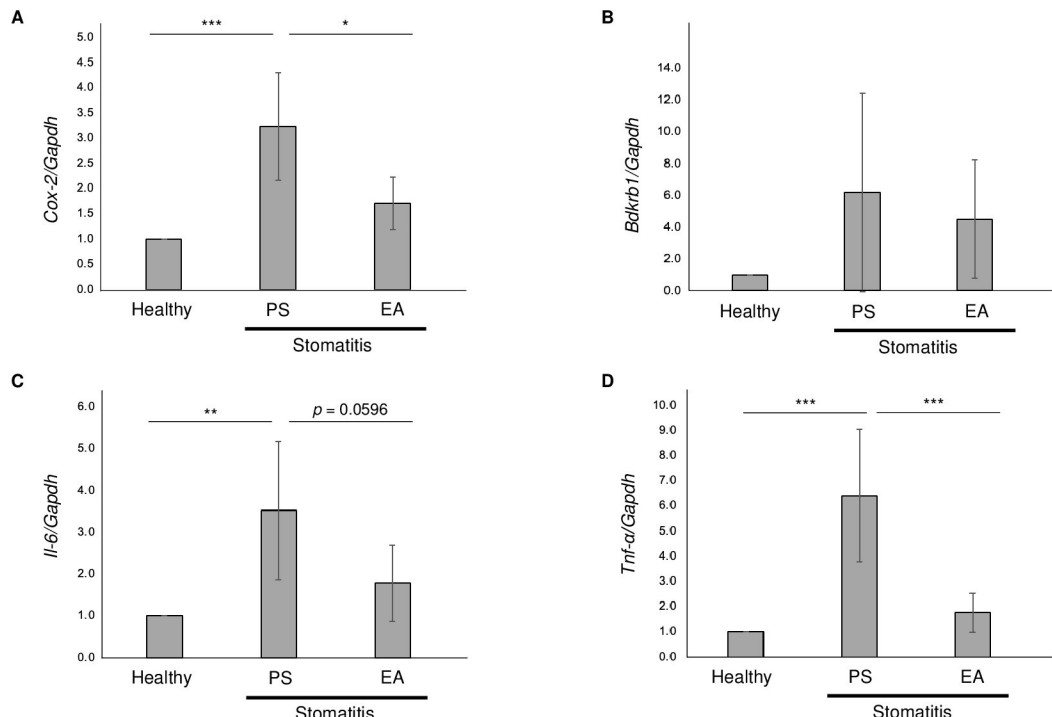

**Fig 5. Gene expression analysis using quantitative polymerase chain reaction.** Acetic acid solution (30 µL) was intramucosally injected into the cheek pouches of hamsters, and 50 µL of EA (15 µg/mL) or PS was immediately administered before and after acetic acid intramucosal injection. After 10 h, the tissue was collected, and total RNA was extracted for qPCR. Healthy: non-stomatitis with the physical saline application, Stomatitis/PS: acetic acid-induced stomatitis with the physical saline application, Stomatitis/EA: acetic acid-induced stomatitis with 15 µg/mL of EA solution application. The internal control was glyceraldehyde 3-phosphate dehydrogenase (*Gapdh*). Data are expressed as the mean ± standard deviation (n = 5 for each group). Tukey–Kramer multiple comparison test, * $p < 0.05$, ** $p < 0.01$, and *** $p < 0.001$.

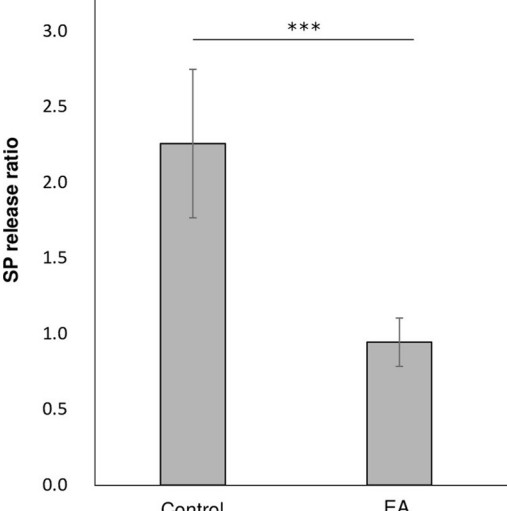

**Fig 6. Substance P release ratio.** The released SP was analyzed using an enzyme-linked immunosorbent assay. The amount of SP released is expressed as the SP release ratio, representing the ratio of evoked SP release to spontaneous SP release. Data are expressed as the mean ± standard deviation (n = 8 for each group). Tukey–Kramer multiple comparison test; *** $p < 0.001$.

Stomatitis can occur anywhere in the mouth, making ointments challenging to apply. However, we considered that a liquid formulation would be appropriate as an analgesic for stomatitis as this formulation can be used to easily and rapidly spread EA throughout the mouth.

EA reportedly promotes wound healing [21,41,42]. In the current study, a significant wound-healing promoting effect was observed by daily applying 1500 μg/mL EA to stomatitis. However, the healing-promoting effect was not confirmed through histopathological or macroscopical examination following a 15 μg/mL EA application. However, both concentrations of EA improved the reduction in weight gain rate caused by eating disorders due to ulcer formation. During their lifetime, male Syrian hamsters gain the most weight from birth to 6 weeks of age. During this period, the rate of weight gain increases linearly if sufficient forage is available. Additionally, hamsters with experimentally induced stomatitis may experience severe pain associated with ulcer stimulation during feeding, resulting in decreased feeding behavior and reduced weight gain. In this study, acetic acid injection induced ulcer formation with severe neutrophil infiltration, and the stomatitis/PS and stomatitis/EA groups had similar cure rates of the wound. However, the stomatitis/EA group demonstrated a higher rate of weight gain at all time points than the stomatitis/PS group. Moreover, we observed a considerable difference in weight between the stomatitis/PS and stomatitis/EA groups on day 7. We examined the analgesic effect of EA and its mechanism of action, which suggests that applying low EA concentrations may have an analgesic effect.

Furthermore, we analyzed the expression of *Tnf-α* and *Il-6* mRNA as the inflammation-related genes and *Cox-2* and *Bdkrb1* mRNA as the pain-related genes in wound tissue through qPCR to determine whether EA suppresses stomatitis pain. The results demonstrated that *Il-6* expression was suppressed. However, *Cox-2* and *Tnf-α* mRNA levels were substantially suppressed in the stomatitis/EA group compared with those in the stomatitis/PS control group in the wound oral area of the stomatitis hamster model (Fig 5).

COX is involved in prostaglandin (PG) production. COX comprises two isozyme isomers: COX-1 and COX-2. COX-1 is a constitutive enzyme [43], primarily distributed in the stomach, kidneys, and platelets [44]. Conversely, monocytes, macrophages, and fibroblasts express COX-2 in response to inflammatory stimulation; therefore, it is referred to as an inducible enzyme [45]. It is a critical enzyme that initiates and promotes inflammatory responses, resulting in tissue injury [46]. Elevated COX-2 expression reduces pain threshold through PG production and, consequently, paves the way for inflammation-related diseases [47]. Pro-inflammatory PGs, which induce hyperalgesia, are produced through COX-2 induction [48,49]; therefore, selective COX-2 inhibitors are widely used as analgesic and anti-inflammatory drugs [50]. In the current study, EA significantly inhibited the elevated COX-2 mRNA expression in the wound area, providing evidence for the analgesic effect of EA.

TNF-α, one of the pro-inflammatory cytokines, is a COX-2 inducer. Therefore, a cross-talk between TNF-α and bradykinin amplifies a protein kinase D phosphorylation cascade, leading to synergistic COX-2 expression *in vitro* [51]. However, in this study, Bdkrb1 mRNA expression levels were not significantly higher in the stomatitis/PS group than in the healthy group; therefore, we could not confirm the effect of EA on Bdkrb1. This should be verified in the future study.

Additionally, we observed that EA considerably inhibited the release of SP from the neuronal cells. SP is an undecapeptide that conveys pain information from the primary sensory neurons to the spinal cord to modulate chronic pain and inflammation [52].

SP stimulates macrophage phagocytic and chemotactic capacity, increasing cytokine, PG E$_2$, and thromboxane B2 production [53]. SP contributes to inflammation development through cytokine production like TNF-α and IL-6 through the mitogen-activated protein kinase and/or nuclear factor kappa B (NF-κB) [54–56].

Fig 7 illustrates the mechanism through which EA suppresses stomatitis pain. Stomatitis increases neuronal-derived SP release, which upregulates downstream pro-inflammatory cytokine expression (TNF-α and IL-6) and increases COX-2 expression, which enhances PG production (Fig 7A). Conversely, EA substantially suppresses SP release, considerably decreasing downstream inflammatory cytokine expression (TNF-α, IL-6), and markedly decreasing COX-2 expression and PG production (Fig 7B). Therefore, EA suppresses stomatitis pain. These EA effects may be attributed to the compounds present in *E. arvense*. For example, quercetin in *E. arvense* promotes wound healing through the extracellular signal-regulated kinase 1/2 mitogen-activated protein kinase and NF-κB pathways [57]. In addition, quercetin inhibits SP expression in the peripheral blood and conjunctival tissue of allergic conjunctivitis mouse models [58]. Moreover, quercetin suppresses COX-2 expression by inhibiting p300 signaling and blocking the binding of multiple transactivators to the COX-2 promoter *in vitro* [59]. Therefore, quercetin or EAs containing large amounts of quercetin may have stronger healing-promoting and/or pain-relieving effects than the EAs used in this study.

Currently, standard analgesics, opioids, and non-steroidal anti-inflammatory drugs are effective treatments for pain; however, several side effects exist. For example, the common side effects of opioid administration include sedation, dizziness, nausea, vomiting, constipation, physical dependence, tolerance, and respiratory depression [60]. Opioids can be considered broad-spectrum analgesic agents that affect numerous organ systems and influence various bodily functions. Additionally, non-steroidal anti-inflammatory drugs are known for several severe side effects, including gastrointestinal toxicities, cardiovascular risks, renal injuries, hepatotoxicity, hypertension, and other minor disorders [61–65]. Furthermore, steroids, commonly used as symptomatic treatments for stomatitis, are easily infectious and can

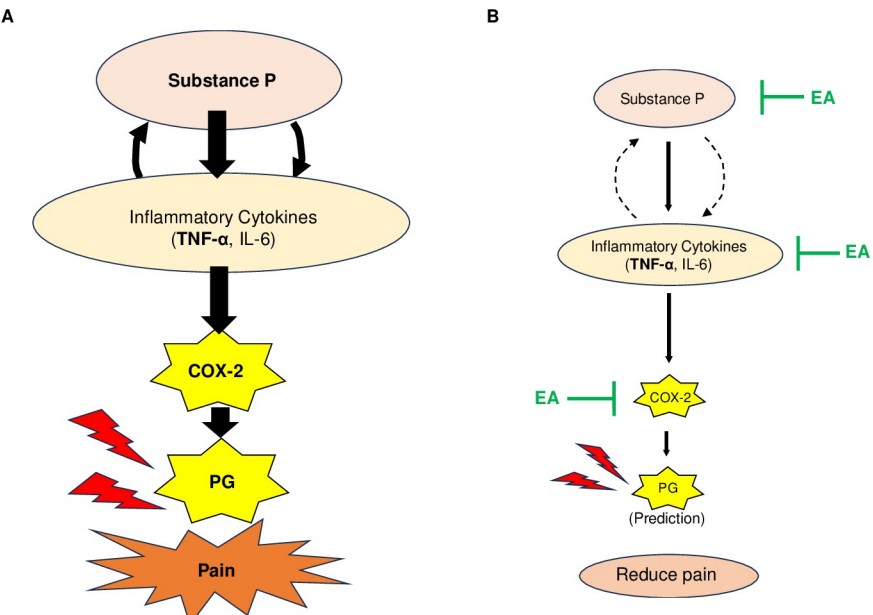

**Fig 7. Proposed *Equisetum arvense* extract (EA) inhibition mechanism of stomatitis pain.** (A) Stomatitis: Enhanced neurotransmitter substance P release from the neurons and increased tumor necrosis factor (TNF)-α and interleukin (IL)-6 expression results in cyclooxygenase-2 (COX-2) expression. Consequently, prostaglandin (PG) is produced, finally resulting in pain development. (B) EA treatment in stomatitis: EA considerably suppresses COX-2 expression, which might be collaborated through the considerable suppression of SP release from the neurons and TNF-α and IL-6 induction in mucosal tissue. Consequently, EA strongly suppresses PG production, resulting in pain reduction.

compromise immunity. Therefore, natural compounds such as EAs, which are extremely safe, could be an alternative strategy to overcome the shortcomings of existing pain treatments.

The limitations of this study include the inability to identify the specific mechanisms of action and active ingredients of EA, as well as the unknown efficacy of EA on human stomatitis. Therefore, future research is necessary to clarify these aspects.

## Conclusions

We demonstrated that EA alleviates pain in acetic acid-induced stomatitis by suppressing inflammatory cytokine induction in the wound area and SP release from the neurons. Therefore, our results suggest that EA may help control the pain associated with stomatitis, such as RAS. In clinical applications, using EA-containing mouthwashes, ointments, or sprays may help control oral mucosal pain with minimal adverse effects and restore the quality of life of patients. However, future studies should clarify the detailed mechanism of EA effects and the effects of human stomatitis.

## Supporting information

**S1 Table. Weight information of animals on d0.** Overall, 52 5-week-old male golden Syrian hamsters (*Mesocricetus auratus*) weighing 95.6 ± 8.4 g (81.6–115.9 g) were used in the study. (XLSX)

**S1 Fig. Timeline of the animal experiment.** Starting from the day after stomatitis induction, 50 μL of EA (15 μg/mL or 1500 μg/mL) or Physiological saline (PS) was applied once daily for seven consecutive days, and the wound area was measured. The hamsters from each group were sacrificed 4th and 7th days after the intramucosal injection of acetic acid solution into the cheek pouch; the cheek pouches were collected, and the tissue sections were prepared for histological evaluation. (TIF)

## Acknowledgments

The authors would like to thank Ayuka Inoue for assistance with the animal experiment.

## Author Contributions

**Conceptualization:** Fumie Shiba, Atsuko Inoue, Koji Ohta, Mutsumi Miyauchi.

**Data curation:** Fumie Shiba, Shiiko Maekawara, Mutsumi Miyauchi.

**Project administration:** Fumie Shiba.

**Writing – original draft:** Fumie Shiba, Mutsumi Miyauchi.

**Writing – review & editing:** Fumie Shiba, Atsuko Inoue, Koji Ohta, Mutsumi Miyauchi.

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
