## [Decision Letter · Decision Letter 0]

30 Jul 2024

PONE-D-24-23705Antinociceptive effect of Equisetum arvense extract on the stomatitis hamster modelPLOS ONE

Dear Dr. Shiba,

Thank you for submitting your manuscript to PLOS ONE. After careful consideration, we feel that it has merit but does not fully meet PLOS ONE’s publication criteria as it currently stands. Therefore, we invite you to submit a revised version of the manuscript that addresses the points raised during the review process. This study project is very interesting and well conducted.

However, it is recommended to delve deeper into some points:Page 3, line 66, it should be anti-osteoclastogenesis.Did authors check the effect of EA on other inflammatory mediators?why Day4’s ratio of [Weight Gain %] of stomatitis/EA vs stomatitis/PS is higher than others? Did authors perform statistical analysis on Day2-6 of stomatitis/EA vs stomatitis/PS?Bdkrb1appeared in the manuscript. Can authors discuss more why Bdkrb1 expression was not affectedTo make this manuscript potential enough and unique in term of “Promotion of Wound Healing” authors should discuss more about how liquid EA application is different from ointment.

We look forward to receiving your revised manuscript.

Kind regards,

Maria Giulia Nosotti, Master's Degree

Academic Editor

PLOS ONE

 [F.S. and M.M. belonged to a collaborative research laboratory of Earth Corporation (https://corp.earth.jp/en/index.html) and Hiroshima University, which used joint research funds to conduct this study.].  

 [F.S. and M.M. belonged to a collaborative research laboratory of Earth Corporation (https://corp.earth.jp/en/index.html) and Hiroshima University, which used joint research funds to conduct this study.].  

5.  We note that Figure(s) 1 and S1 in your submission contain copyrighted images. All PLOS content is published under the Creative Commons Attribution License (CC BY 4.0), which means that the manuscript, images, and Supporting Information files will be freely available online, and any third party is permitted to access, download, copy, distribute, and use these materials in any way, even commercially, with proper attribution. For more information, see our copyright guidelines: http://journals.plos.org/plosone/s/licenses-and-copyright.

a. You may seek permission from the original copyright holder of Figure(s) 1 and S1 to publish the content specifically under the CC BY 4.0 license. 

6. Please include captions for all your Supporting Information files at the end of your manuscript, and update any in-text citations to match accordingly. Please see our Supporting Information guidelines for more information: http://journals.plos.org/plosone/s/supporting-information. 

Reviewers' comments:

Reviewer's Responses to Questions

**Comments to the Author**

1. Is the manuscript technically sound, and do the data support the conclusions?

Reviewer #1: Yes

Reviewer #2: Yes

Reviewer #3: Yes

2. Has the statistical analysis been performed appropriately and rigorously? 

Reviewer #1: Yes

Reviewer #2: Yes

Reviewer #3: Yes

3. Have the authors made all data underlying the findings in their manuscript fully available?

Reviewer #1: Yes

Reviewer #2: Yes

Reviewer #3: Yes

4. Is the manuscript presented in an intelligible fashion and written in standard English?

Reviewer #1: Yes

Reviewer #2: Yes

Reviewer #3: Yes

5. Review Comments to the Author

Reviewer #1: The study design was appropriate to answer the research question (including the use of appropriate controls), and the conclusions supported by the evidence were presented.

The methods were sufficiently described to allow the study to be repeated.

The use of statistics and treatment of uncertainties were appropriate

The presentation of the work was clear.

Reviewer #2: The manuscript entitled "Antinociceptive effect of Equisetum arvense extract on the stomatitis hamster model" was reviewed. Please respond to the questions and perform some corrections or changes suggested as follows

1. Why did you use Equisetum arvense extract especially for this research although there are many other medicinal plant extracts or synthetic materials with the same properties?

2. Why did you use the hamster model instead of the rat or mouse model?

3. Did you stain the pathological sections with histochemical stains such as Van Gieson or Masson's trichrome to compare collagen fibers in the groups?

4. Some figures have no scale number in the histopathologic figure (figure 2). That's better the number of the scales to be written in black color.

5. Considering each plant's extract consists of different components, do you think which EA extract especially has the potential ability to reduce pain and promote wound healing? Accordingly, do you think it is not better to purify the main components of this herbal extract and use the more useful components of this extract?

6. Why didn't you use different concentrations of EA in this research comparatively?

7. Considering that in the first days of wound formation, cells with neutrophils predominate and given the source of Cox 2 production is mostly monocytes, macrophages, and fibroblasts, don't you think measuring Cox 2 on days 4 and 7 in addition to 10 h after creating wound could be useful for a more accurate evaluation? However, pain reduction in the first 24 hours can be important for feeding.

8. What is the unit of SP content that you measured by ELISA? You mentioned the ratio in figure 5.

9. The measuring time of SP content is not mentioned in the text. SP modulates chronic pain and inflammation. So why did you measure it for acute pain? Wouldn't it be better if you measured and compared it on different days?

Reviewer #3: Shiba F et al., showed the interesting study on the “Antinociceptive effect of Equisetum arvense extract on the stomatitis hamster model”. The results are drastically significant and study designs are interesting; however, authors need some additional modifications. Some concerns below needs to be addressed.

1- Page 3, line 66, it should be anti-osteoclastogenesis.

2- Did authors check the effect of EA on other inflammatory mediators?

3- Figure 3, why Day4’s ratio of [Weight Gain %] of stomatitis/EA vs stomatitis/PS is higher than others? Did authors perform statistical analysis on Day2-6 of stomatitis/EA vs stomatitis/PS?

4- Bdkrb1appeared in the manuscript. Can authors discuss more why Bdkrb1 expression was not affected?

5- To make this manuscript potential enough and unique in term of “Promotion of Wound Healing” authors should discuss more about how liquid EA application is different from ointment.

6. PLOS authors have the option to publish the peer review history of their article (what does this mean?). If published, this will include your full peer review and any attached files.

Reviewer #1: **Yes: **Rahim Mohammadi

Reviewer #2: No

Reviewer #3: **Yes: **Chanbora Chea

---

## [Author Response · Author response to Decision Letter 0]

28 Oct 2024

We were unable to include everything in this section, so please refer to the Response to Reviewers for more details.

---

## [Editor Report · Decision Letter 1]

31 Oct 2024

Antinociceptive effect of Equisetum arvense extract on the stomatitis hamster model

PONE-D-24-23705R1

Dear Dr. Shiba,

We’re pleased to inform you that your manuscript has been judged scientifically suitable for publication and will be formally accepted for publication once it meets all outstanding technical requirements.

Kind regards,

Maria Giulia Nosotti, Master's Degree

Academic Editor

PLOS ONE
---

## [Editor Report · Acceptance letter]

11 Nov 2024

PONE-D-24-23705R1 

PLOS ONE

Dear Dr. Shiba, 

I'm pleased to inform you that your manuscript has been deemed suitable for publication in PLOS ONE. Congratulations! Your manuscript is now being handed over to our production team.

Kind regards, 

on behalf of

Dr. Maria Giulia Nosotti 

Academic Editor

PLOS ONE